# 4D TENSOR MULTI-TASK CONTINUAL LEARNING FOR DISEASE DYNAMIC PREDICTION

## ABSTRACT

Machine learning techniques for predicting Alzheimer's disease (AD) progression can substantially help researchers and clinicians establish strong AD preventive and treatment strategies. However, current research on AD prediction algorithms encounters challenges with monotonic data form, small dataset and scarcity of time-continuous data. To address all three of these problems at once, we propose a novel machine learning approach that implements the 4D tensor multi-task continual learning algorithm to predict AD progression by quantifying multi-dimensional information on brain structural variation and knowledge sharing between patients. To meet real-world application scenarios, the method can integrate knowledge from all available data as patient data increases to continuously update and optimise prediction results. To evaluate the performance of the proposed approach, we conducted extensive experiments utilising data from the Alzheimer's Disease Neuroimaging Initiative (ADNI). The results demonstrate that the proposed approach has superior accuracy and stability in predicting various cognitive scores of AD progression compared to single-task learning, benchmark and state-of-the-art multi-task regression methods. The proposed approach identifies structural brain variations in patients and utilises it to accurately predict and diagnose AD progression from magnetic resonance imaging (MRI) data alone, and the performance of the model improves as the MRI data increases.

## 1 INTRODUCTION

Alzheimer's disease is a severe neurodegenerative condition in which neurons and their connections degrade over time, resulting in a wide spectrum of dementia symptoms including cognitive impairment, memory loss, and executive dysfunction (Khachaturian, 1985). Standard AD prediction methods focus on discovering and identifying important biomarkers from a variety of modalities and then learning the model as a regression problem to calculate cognitive scores at various time periods. Existing prediction models for AD progression include classic machine learning regression techniques (Tabarestani et al., 2020; Wang et al., 2018), deep learning methods based on neural networks (Liu et al., 2014; Nguyen et al., 2018), and survival models based on statistical probabilities (Doody et al., 2010; Fenn and Gray, 2012).

There are three main problems with the above model. The first is the problem of small dataset. Data on neurological diseases such as AD are difficult to obtain. The second is the scarcity of time-continuous data which means as the disease progresses, the number of available datasets will further decrease. Traditional machine learning techniques have limited accuracy, and small datasets make it difficult to construct deep learning models with superior precision. The third problem is the monotonic form of the data. The input features of the above model are represented as second-order matrices containing patient and biomarker dimensions, which makes it challenging to predict and analyse disease progression from various dimensions. Meanwhile, as the second-order matrix can only focus on a single biomarker, correlation knowledge between different AD biomarkers will be lost.

For the monotonic data form problem, this research attempts to build an AD prediction model utilising a third-order tensor to better describe diverse aspects of AD data through both spatial and temporal dimensions. Tensors can be utilised in regression algorithms to enhance prediction accuracy, stability and interpretability by better representing AD biomarker features. Figure 1 illustrates the 3D tensor

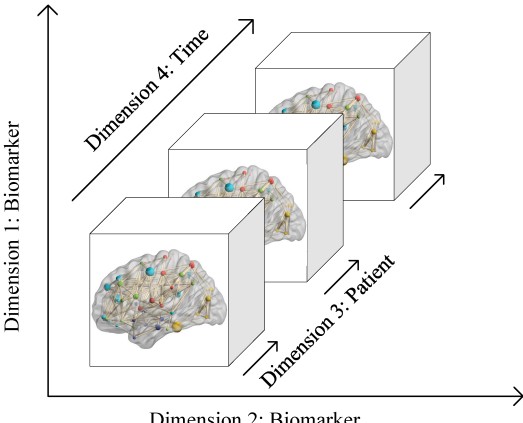

Figure 1: The 4D tensor data structure constructed and utilised in the research.

constructed and utilised in the research, and the 4D tensor with the time dimension introduced as will be mentioned later.

For the small dataset problem, multi-task learning (MTL) can share knowledge and information between tasks, outperforms standard single-task learning approaches in terms of generalizability, interpretability, and prediction accuracy, and is most efficient when sample numbers are small (Zhou et al., 2013). As a result, we utilise a tensor-based MTL technique to include spatio-temporal information on structural variations in the brain to predict the progression of AD. Specifically, we first utilise a similarity-computation-based approach to simultaneously quantify the magnitude and direction information of structural variations in the brain, the method characterises the similarity of morphological variation trends between different biomarkers as a third-order tensor with dimensions corresponding to the first biomarker, the second biomarker and the patient sample. The proposed algorithm then performs a CANDECOMP/PARAFAC (CP) symmetric decomposition of the tensor (Kolda and Bader, 2009) and extracts a set of rank-one latent factors from the data. The predictions for each patient sample (the task in this research) share these latent factors.

In the real world, patients suspected of AD will continue to go to hospital for testing. Subsequent incremental data is wasted if only a baseline model is utilised or if consecutive test records of patients cannot be reasonably integrated. To solve this problem, we apply the concept of continuous learning to our approach, which can update the prediction results by allowing the model to receive new MRI data while receiving all the latent factors from all previous prediction models. Figure 2 depicts the architecture, learning process and real-world applications of the proposed approach.

The following are the primary contributions of this work:

- We present a novel approach to AD progression prediction that requires only MRI data to provide accurate predictions, which utilises a tensor-based MTL algorithm to seamlessly integrate and share spatio-temporal information based on brain structural variations and the latent factors of its biomarkers, thereby significantly improving the accuracy and stability of AD progression prediction under the problems of small data sets and monotonic data formats.

- For real-world applications, the presented AD dynamic prediction utilises the conception of continual learning to concurrently acquire knowledge of all previous prediction models in order to update prediction results while receiving new MRI data for prediction. Experimental results reveal that prediction accuracy improves continually as the number of MRI detections increases.

- We identified and analysed important relative structural variation correlations between brain biomarkers in the prediction of AD progression, which could be utilised as potential indicators for early identification of AD.

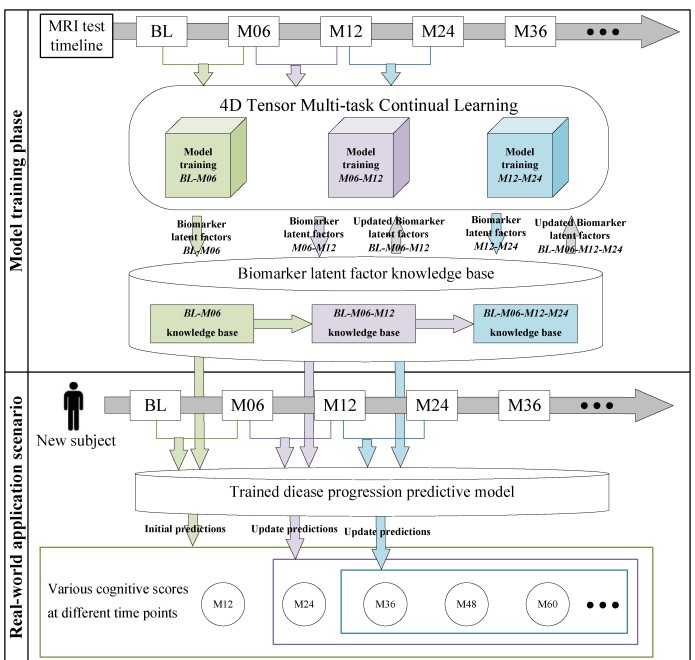

Figure 2: Architecture, learning procedure and real-world application for the proposed 4D tensor multi-task continual learning approach. From a continuous prediction perspective, new prediction model acquires all the latent factors from previous models and updates the predictions whenever the patient's MRI data is updated. (The notation "BL" denotes date of the patient's first admission for screening, "M06" denotes the time point 6 months after the first visit, "M12" denotes the time point 12 months after the first visit, etc.).

## 2 RELATED WORK

Numerous studies in brain science have focused on the distinctions in brain structure between CN (cognitively normal), MCI (mild cognitive impairment) and AD. (Thompson et al., 2004; Vemuri et al., 2009; Singh et al., 2006) integrate imaging data from large human populations to discover patterns of brain structure and function related with Alzheimer's disease, normal ageing, schizophrenia, and abnormal brain development. On this basis, the correlation between MRI biomarkers of AD is also a major focus of brain science research. (Wee et al., 2013; He et al., 2008) enhanced classification performance of AD and its precursor stages by merging relevant information with ROI-based data and correlating regional mean cortical thickness. Abovementioned study found differences in brain biomarkers for CN, MCI and AD. It also examined and analysed relationships between AD progression and biomarkers. The preceding researches solely focuses on a single biomarker or a single category of biomarkers, neglecting the connection and correlation of spatio-temporal variation operating in different categories of biomarkers, which is critical for depicting AD symptoms.

Multi-task learning attempts to jointly learn numerous related tasks to ensure that the knowledge contained in one task can be applied by other tasks, ultimately boosting the generalisation performance of all tasks (Zhang and Qiang, 2021). MTL technology is extensively implemented in the biomedical engineering field, for our research case AD, MTL provides a wide range of applications in numerous domains. In terms of feature selection approach, (Zhang and Shen, 2012; Zeng et al., 2021) presented multi-task feature selection techniques which evaluate internal connection between several related tasks, and chooses feature sets relevant to all tasks. In terms of feature learning approach, existing approaches have focused on modelling task interactions through use of novel regularisation techniques (Jiang et al., 2018; Wang et al., 2019; Cao et al., 2018; Peng et al., 2019). In terms of low-rank approach, (Chen et al., 2011) presented a robust multi-task learning method that employs a low-rank structure to preserve task connections while identifying anomalous tasks using a group sparse structure. In contrast to above approach, we assumed that knowledge sharing between prediction

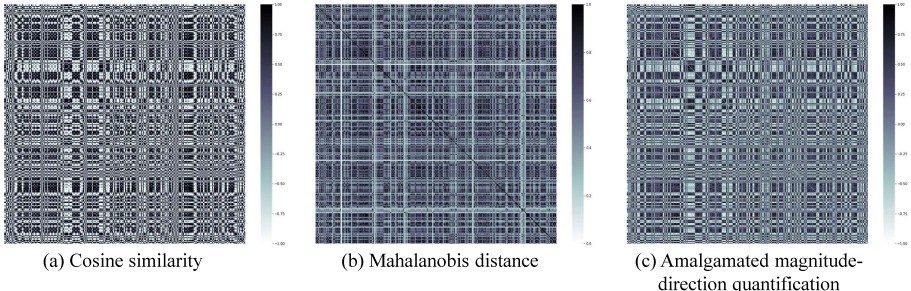

(a) Cosine similarity          (b) Mahalanobis distance          (c) Amalgamated magnitude-direction quantification

Figure 3: For quantifying brain structural variation, compare instances of correlation matrix distributions for (a) Cosine similarity, (b) Mahalanobis distance, and (c) Amalgamated magnitude-direction quantification. (The scale for (b) Mahalanobis distance from top to bottom is 1.0, 0.8, 0.6, 0.4, 0.2, 0.0. The scale for (a) Cosine similarity and (c) Amalgamated magnitude-direction quantification from top to bottom is 1.00, 0.75, 0.50, 0.25, 0.00, -0.25, -0.50, -0.75, -1.00.)

tasks for different patients is expected to improve achievable performance, and therefore we set up prediction task for a single patient as a task, which is a small-scale manner of task setting.

# 3 METHODOLOGY

## 3.1 DENOTATION

For brevity, we represent tensors as italic capital letters, such as $X$ or $Y$, and matrices by capital letters, such as A or B. Vectors are denoted by lowercase letters such as x whereas Scalars are denoted by italic lowercase letters such as $a$.

## 3.2 AMALGAMATED MAGNITUDE-DIRECTION QUANTIFICATION FOR BRAIN STRUCTURE VARIATION

The correlation of structural variance between different brain biomarkers was calculated utilising two consecutive MRI scans and this work expands and executes it throughout a number of following time periods (BL to M06, M06 to M12, M12 to M24). For instance, we calculated the rate of change and velocity for each brain biomarker utilising MRI at the time points BL and M06. The rate of change is $\frac{x_{M06}-x_{BL}}{x_{BL}}$, the velocity is $\frac{x_{M06}-x_{BL}}{t_{M06}-t_{BL}}$ per month, where $x$ is the test value of brain biomarkers and $t$ is the MRI detection dates. The rate of change and velocity were then utilised to construct a vector representing the structural variation trend of the brain biomarker.

Then we present a two-stage qualitative method for the simultaneous assessment of information on the magnitude and direction of structural variation between different brain biomarkers. Firstly, we utilised the Mahalanobis distance to calculate the similarity of the absolute values of two vectors to extract information on the magnitude of the structural variation correlation between two MRI biomarkers. The Mahalanobis distance is utilised because it divides the covariance matrix in the formula and makes the results scale independent. The Mahalanobis distance between the absolute values of vectors $x_i$ and $x_j$ is stated as: $\mathrm{Ma}\left(|x_i|,|x_j|\right) = \sqrt{\left(|x_i|-|x_j|\right)^{\mathrm{T}} \mathrm{S}^{-1}\left(|x_i|-|x_j|\right)}$, where S is covariance matrix. The quantified Mahalanobis distance ranges from 1 to 0, where 1 indicates complete similarity and 0 indicates complete dissimilarity. Secondly, the directional information is concatenated to the values. We observed that the structural variation directional correlation between two brain biomarkers existed in following five cases: 1) both grow, 2) both decline, 3) one grows and the other declines, 4) one changes and the other does not change, and 5) both remain unchanged. We indicate cases 1) and 2) as synchronous variations, case 3) as asynchronous variations, and cases 4) and 5) as completely unrelated variations. To incorporate directional information, the numbers previously obtained utilising the Mahalanobis distance are mapped to values between 1 and -1 utilising a mapping function (1). Where 1 means completely relevant in the case of synchronous variation, 0

means completely irrelevant and -1 means completely relevant in the case of asynchronous variation.

$$\begin{cases} \text{x} = \text{x, if two biomarkers varied synchronously} \\ \text{x} = -\text{x, if two biomarkers varied asynchronously} \\ \quad \text{x} = 0, \text{if two biomarkers are not relevant} \end{cases} \tag{1}$$

Figure 3 demonstrates the ability of our proposed approach to simultaneously capture information on the magnitude and direction of structural variation in the brain. Cosine similarity and Mahalanobis distance are classical similarity calculation methods, which utilise the direction and magnitude information of vectors as criteria for calculation respectively, whereas our approach captures information on both magnitude and direction, resulting in a matrix distribution similar to that of cosine similarity, but with the same smooth data distribution and diverse data characteristics as the Mahalanobis distance. It enables the AD progression prediction approach to incorporate greater detailed information on brain structural variation while improving the interpretability for brain biomarker correlations in AD progression during results analysis process.

### 3.3 TENSOR MULTI-TASK CONTINUAL LEARNING

To predict various cognitive scores (e.g., MMSE and ADAS-Cog) for AD at future time points. Consider a multi-dimensional tensor multi-task continual regression problem for $t$ time points, $n$ training samples with $d_1$ and $d_2$ features. Let $X \in \mathbb{R}^{d_1 \times d_2 \times n}$ be the input three-dimensional tensor from two successive MRI records and it is the combination of correlation matrix for all $n$ samples, $X_n \in \mathbb{R}^{d_1 \times d_2}$, $Y = [y_1, \cdots, y_t] \in \mathbb{R}^{n \times t}$ be the targets (clinical scores) and $y_t = [y_1, \cdots, y_n] \in \mathbb{R}^n$ is the corresponding target at various time points.

For $t$-th prediction time point, the objective function of the proposed approach can be stated as follows:

$$L_t(X, \text{y}_t) = \min_{\text{W}_t, \text{A}_t, \text{B}_t, \text{C}_t} \frac{1}{2} \|\hat{\text{y}}_t - \text{y}_t\|_2^2 + \frac{\lambda}{2} \|X - [\![\text{A}_t, \text{B}_t, \text{C}_t]\!]_\text{S}\|_\text{F}^2 + \beta \|\text{W}_t, \text{A}_t, \text{B}_t, \text{C}_t\|_1$$

$$\hat{y}_n = \sum_{i=1}^{d_1} \sum_{j=1}^{d_2} \text{U}_{ij}, \tag{2}$$

$$\text{where } \text{U} = \left(\eta \odot \text{V} + (1 - \eta) \odot (\text{A}_t \text{B}_t^\text{T})\right) \odot \text{K} \odot \text{W}_t \odot \text{X}_n, \text{U} \in \mathbb{R}^{d_1 \times d_2}.$$

where the first term calculates the empirical error with training data, $\hat{\text{y}}_t = [\hat{y}_1, \cdots, \hat{y}_n] \in \mathbb{R}^n$ are predicted values, $\text{A}_t \in \mathbb{R}^{d_1 \times r}$ is latent factor matrix for the first biomarker dimension and $\text{B}_t \in \mathbb{R}^{d_2 \times r}$ is latent factor matrix for the second biomarker dimension with $r$ latent factors, $\text{W}_t \in \mathbb{R}^{d_1 \times d_2}$ is model parameter matrix for the $t$-th prediction time point, $\lambda$ and $\beta$ are regularization parameters. $\text{V} \in \mathbb{R}^{d_1 \times d_2}$ is knowledge base matrix which stores principal biomarker latent factors from all preceding model predictions. $\text{V}$ is updated after each model prediction with following equation: $\text{V}_{new} = \eta \odot \text{V}_{old} + (1 - \eta) \odot (\text{A}_t \text{B}_t^\text{T})$. The hyperparameter $\eta$ is utilised to control the proportion of preceding and present knowledge base that is employed. Acquiring latent factors by optimising the symmetric CP tensor decomposition objective function $\|X - [\![\text{A}_t, \text{B}_t, \text{C}_t]\!]_\text{S}\|_\text{F}^2$, where $X = [\![\text{A}_t, \text{B}_t, \text{C}_t]\!]_\text{S} = \sum_{i=1}^{r} \frac{1}{2}(\text{a}_i^t \circ \text{b}_i^t \circ \text{c}_i^t + \text{b}_i^t \circ \text{a}_i^t \circ \text{c}_i^t)$ and $\circ$ denote outer product operation between two vectors, while $\text{a}_i^t$, $\text{b}_i^t$ and $\text{c}_i^t$ correspond to vectors related with $i$-th latent factor for $t$-th prediction time point. $\|\text{W}_t, \text{A}_t, \text{B}_t, \text{C}_t\|_1$ applying the $\ell1$-norm on the $\text{W}_t$, $\text{A}_t$, $\text{B}_t$ and $\text{C}_t$ matrices individually. We utilise the operator $\odot$ as follows: $\text{Z} = \text{M} \odot \text{N}$ denotes $z_{ij} = m_{ij}n_{ij}$, for all $i, j$. And $\text{Z} = \text{m} \odot \text{N}$ denotes $z_{ij} = mn_{ij}$, for all $i, j$. The matrix $\text{K} \in \mathbb{R}^{d_1 \times d_2}$ is the duplicate data correction matrix which was implemented to fix the duplicate data problem because the correlation tensor for brain structural variation created by the proposed quantification approach is a symmetric tensor, which means that the correlations between biomarkers are calculated in pairs, resulting in half of the data being duplicates. It is stated as follows:

$$\text{K} = \begin{bmatrix} 0 & 1 & \cdots & 1 \\ \vdots & \ddots & & \vdots \\ & & & 1 \\ 0 & \cdots & & 0 \end{bmatrix} \in \mathbb{R}^{d_1 \times d_2} \tag{3}$$

For all prediction time points together, the objective function can be stated as follows:

$$L(X, \text{Y}) = \min_{\text{W}_f} \sum_1^t L_t(X, \text{y}_t) + \theta \| \text{W}_f \text{P}(\alpha) \|_\text{F}^2 \tag{4}$$

Table 1: Demographic characteristic of the studied subjects valued are specified as mean±standard deviation.

| TIME POINT | ATTRIBUTE | MMSE | ADAS-COG |
|---|---|---|---|
| M12 | SAMPLE SIZE (CN, MCI, AD) | 1334 (359, 726, 249) | 1321 (354, 722, 245) |
| | GENDER(F/M) | 580/754 | 575/746 |
| | AGE | 74.9±7.2 | 74.9±7.1 |
| M24 | SAMPLE SIZE (CN, MCI, AD) | 1127 (335, 620, 172) | 1105 (332, 613, 160) |
| | GENDER(F/M) | 493/634 | 481/624 |
| | AGE | 75.8±7.1 | 75.8±7.1 |
| M36 | SAMPLE SIZE (CN, MCI, AD) | 745 (206, 528, 11) | 730 (203, 518, 9) |
| | GENDER(F/M) | 324/421 | 318/412 |
| | AGE | 76.4±7.0 | 76.4±7.1 |
| M48 | SAMPLE SIZE (CN, MCI, AD) | 585 (218, 360, 7) | 579 (215, 357, 7) |
| | GENDER(F/M) | 259/326 | 261/318 |
| | AGE | 76.9±6.8 | 77.0±6.8 |
| M60 | SAMPLE SIZE (CN, MCI, AD) | 333 (115, 216, 2) | 330 (115, 213, 2) |
| | GENDER(F/M) | 144/189 | 143/187 |
| | AGE | 78.2±6.7 | 78.3±6.7 |

where $\| W_f P(\alpha) \|_F^2$ is the generalized temporal correlation term, model parameter matrix $W_f \in \mathbb{R}^{(d_1 \times d_2) \times t}$ is the temporal dimension unfolding of model parameter tensor $W \in \mathbb{R}^{d_1 \times d_2 \times t}$, $\theta$ is the regularization parameter. The generalised temporal correlation states that while diagnosing AD, the expert analyses not only the patient's present symptoms, but also their previous symptoms. As a result, we can formulate the more realistic temporal correlation assumption utilising matrix multiplication formulation:

$$WP(\alpha) = WHD_1(\alpha_1) D_2(\alpha_2) \cdots D_{t-2}(\alpha_{t-2}) \tag{5}$$

where $P(\alpha)$ denotes the correlation between disease progresses, it involves hyperparameters $\alpha$, which depict the relational degree between the present progression and all previous progressions. The relational degree criteria differ for each stage of disease progression because each stage's impact on the stage after it may not always be constant and it depends on the outcome of cross-validation. $H \in \mathbb{R}^{t \times (t-1)}$ has the following definition: $H_{ij} = 1$ if $i = j$, $H_{ij} = -1$ if $i = j + 1$ and $H_{ij} = 0$ otherwise. $D_i(\alpha_i) \in \mathbb{R}^{(t-1) \times (t-1)}$ is an identity matrix and the value of $D_{i_{m,n}}(\alpha_i)$ is substituted by $\alpha_i$ if $m = i$, $n = i + 1$, the value of $D_{i_{m,n}}(\alpha_i)$ is substituted by $1 - \alpha_i$ if $m = n = i + 1$.

Latent factors $A \in \mathbb{R}^{d_1 \times r \times t}$, $B \in \mathbb{R}^{d_2 \times r \times t}$, $C \in \mathbb{R}^{n \times r \times t}$ and model parameter $W \in \mathbb{R}^{d_1 \times d_2 \times t}$ can be trained by optimising the objective function for each group of variables to be resolved consecutively. Since not all parts of the objective function are differentiable, we use proximal gradient descent to solve each subproblem. Our objective function's components related to Frobenius norms are differentiable, but those related to the $\ell 1$-norms that ensure sparsity are not. Proximal approach is frequently utilised to construct the proximal problems for a non-smooth objective function (Gong et al., 2014; Han and Zhang, 2015). The strategy can make the design of distributed optimization algorithms simpler and accelerate the convergence of the optimization process.

## 4 EXPERIMENTAL SETTINGS

### 4.1 DATASET

The Alzheimer's Disease Neuroimaging Initiative (ADNI) database (adni.loni.usc.edu) provided the data required to construct this paper. The FreeSurfer image analysis system (http://surfer.nmr.mgh.harvard.edu/) was utilised to perform volumetric segmentations and cortical reconstruction utilising imaging data from the ADNI database, which contains all ADNI subprojects (ADNI 1, 2, GO, 3). We obtained MRI data from the ADNI database and proceeded with the pre-processing steps listed below: 1) Image records having failed quality control are removed; 2) Participants who lacked BL and M06 MRIs were eliminated; 3) Remove features that have more than half of their values missing; 4) The average of the features was used to fill in missing data; 5) Individuals with no follow-up MRI detections for AD dynamic prediction are excluded.

After the pre-processing procedure, a total of 313 MRI features were obtained and can be classified into the following five categories: average cortical thickness (TA), standard deviation in cortical thickness (TS), the total surface area of the cortex (SA), the volumes of specific white matter

Table 2: Comparison of the results from our proposed methods with benchmarks and state-of-the-art methods for MMSE at time points M12 to M60. The best results are bolded.

| TARGET: MMSE | INPUT MRI DATA | M12 RMSE | M24 RMSE | M36 RMSE | M48 RMSE | M60 RMSE |
|---|---|---|---|---|---|---|
| LASSO | BL, M06 | 2.0189±0.1243 | 2.5647±0.5361 | 3.5177±0.4846 | 3.9029±1.0973 | 3.8909±0.3210 |
| | BL, M06, M12 | - | 2.2291±0.2059 | 2.7919±0.4040 | 3.6100±0.6652 | 4.0934±1.3271 |
| | BL, M06, M12, M24 | - | - | 2.5759±0.4625 | 3.9002±0.8624 | 3.5580±0.5912 |
| cFSGL | BL, M06 | 1.5432±0.1361 | 1.6363±0.3951 | 1.5900±0.1767 | 2.1683±0.1951 | 2.7854±0.4179 |
| | BL, M06, M12 | - | 1.5515±0.2360 | 1.7764±0.2443 | 1.8191±0.4650 | 2.5032±0.7737 |
| | BL, M06, M12, M24 | - | - | 1.5377±0.1716 | 1.9078±0.5090 | 2.0477±0.3496 |
| NC-CMTL | BL, M06 | 1.8486±0.5758 | 1.7701±0.2656 | 1.9232±0.1460 | 2.5262±0.1845 | 3.5791±0.3093 |
| | BL, M06, M12 | - | 1.6317±0.2441 | 2.1279±0.3990 | 2.6328±0.2762 | 3.4193±0.7643 |
| | BL, M06, M12, M24 | - | - | 1.8975±0.2608 | 2.6317±0.5001 | 3.2897±0.3539 |
| FL-SGL | BL, M06 | 1.8711±0.2819 | 1.8183±0.1100 | 1.8818±0.3331 | 2.8564±0.6854 | 3.2593±1.0596 |
| | BL, M06, M12 | - | 2.2575±0.2883 | 1.7402±0.5217 | 2.8533±0.2367 | 4.5144±1.9577 |
| | BL, M06, M12, M24 | - | - | 1.8922±0.1033 | 2.3626±0.1123 | 3.8178±0.4960 |
| GAMTL | BL, M06 | 1.4821±0.2615 | 1.5014±0.1068 | 1.8501±0.1367 | 2.3420±0.1378 | 3.5989±0.2666 |
| | BL, M06, M12 | - | 1.5728±0.2401 | 1.4358±0.1723 | 1.9748±0.1177 | 2.6538±0.5120 |
| | BL, M06, M12, M24 | - | - | 1.7845±0.1249 | 2.0169±0.5532 | 3.5322±0.5704 |
| 4DTMTCL | BL, M06 | **1.3554±0.1033** | 1.3898±0.0881 | 1.4051±0.0843 | 1.5140±0.0415 | 2.0128±0.6371 |
| | BL, M06, M12 | - | **1.3744±0.0962** | 1.5025±0.1802 | 1.4790±0.0429 | 1.9820±0.5961 |
| | BL, M06, M12, M24 | - | - | **1.3892±0.0811** | **1.4548±0.0693** | **1.8849±0.4703** |

Table 3: Comparison of the results from our proposed methods with benchmarks and state-of-the-art methods for ADAS-Cog at time points M12 to M60. The best results are bolded.

| TARGET: ADAS-COG | INPUT MRI DATA | M12 RMSE | M24 RMSE | M36 RMSE | M48 RMSE | M60 RMSE |
|---|---|---|---|---|---|---|
| LASSO | BL, M06 | 5.6398±0.2601 | 6.0893±1.2338 | 7.1638±2.0246 | 9.2744±1.7223 | 11.5160±1.8211 |
| | BL, M06, M12 | - | 7.8403±1.6240 | 7.6791±1.3058 | 9.2590±1.9429 | 12.6903±1.5302 |
| | BL, M06, M12, M24 | - | - | 8.2674±1.8710 | 8.1798±1.8091 | 10.8189±0.9235 |
| cFSGL | BL, M06 | 3.7759±0.4832 | 2.8756±0.6098 | 3.6017±0.7752 | 5.5702±2.2033 | 6.7801±2.5726 |
| | BL, M06, M12 | - | 3.7889±0.8487 | 3.5463±0.2307 | 3.9609±0.7503 | 5.6611±0.9303 |
| | BL, M06, M12, M24 | - | - | 4.1155±0.6206 | 3.4266±0.5278 | 5.7447±0.9492 |
| NC-CMTL | BL, M06 | 3.9772±0.6216 | 3.7622±0.6284 | 3.5110±0.2063 | 4.9121±1.9753 | 5.7035±1.6713 |
| | BL, M06, M12 | - | 4.3042±0.2030 | 3.4703±0.4813 | 3.9763±0.5041 | 6.6742±1.9385 |
| | BL, M06, M12, M24 | - | - | 4.0753±0.7950 | 5.2504±1.7646 | 5.1796±1.1334 |
| FL-SGL | BL, M06 | 5.9684±0.1981 | 6.2061±0.9391 | 6.3641±0.6227 | 7.9175±1.3813 | 11.1045±1.2027 |
| | BL, M06, M12 | - | 4.5405±1.0258 | 5.5387±0.6943 | 6.4337±1.1745 | 8.0628±2.1986 |
| | BL, M06, M12, M24 | - | - | 5.2439±1.6629 | 6.7413±1.8830 | 6.9925±1.4413 |
| GAMTL | BL, M06 | 3.9972±0.8691 | 3.5136±0.1790 | 4.3254±0.2420 | 3.9275±0.6892 | 5.5146±0.3708 |
| | BL, M06, M12 | - | 4.2089±0.2129 | 4.1534±0.5414 | 4.5681±1.1013 | 6.7227±1.6742 |
| | BL, M06, M12, M24 | - | - | 3.1704±1.0534 | 4.3256±0.4681 | 5.2752±1.4584 |
| 4DTMTCL | BL, M06 | **1.3831±0.0743** | 1.5662±0.1601 | 1.5314±0.1488 | 1.4487±0.2495 | 2.2031±0.1996 |
| | BL, M06, M12 | - | **1.5573±0.1695** | 1.4293±0.2843 | 1.4625±0.2598 | 2.1739±0.1591 |
| | BL, M06, M12, M24 | - | - | 1.5735±0.1004 | **1.3356±0.2183** | **2.1201±0.0862** |

parcellations (SV) and the volumes of cortical parcellations (CV). Table 1 illustrates the demographic characteristics of the ADNI MRI data used in this research.

## 4.2 EVALUATION METRICS

The tensor multi-task continual model was constructed utilising the correlation tensor of structural variation trends between MRI brain biomarkers. The data was randomly split into a training set and a test set in a ratio of 9:1. As the value of model parameters ($\lambda$, $\beta$ and $\theta$), hyperparameters ($\alpha$ and $\eta$) and latent factor $r$ must be stated during training phase, we utilise the 5-fold cross-validation with training data. The research evaluates the performance of various methods in terms of AD prediction at each time point, with the root mean square error (rMSE) working as the critical evaluation metric. We utilise normalised mean square error (nMSE) for the overall regression performance metrics, which is widely used in multi-task learning research (Argyriou et al., 2008). The rMSE and nMSE measurements are as follows:

$$\text{rMSE}\,(\text{y}, \hat{\text{y}}) = \sqrt{\frac{\|\text{y} - \hat{\text{y}}\|_2^2}{n}} \tag{6}$$

$$\text{nMSE}\left(\text{Y}, \hat{\text{Y}}\right) = \frac{\sum_{i=1}^{t} \| \text{Y}_i -, \hat{\text{Y}}_i \|_2^2 \Big/ \sigma\,(\text{Y}_i)}{\sum_{i=1}^{t} n_i} \tag{7}$$

where for the rMSE, y is ground truth of the target at a single time point and $\hat{\text{y}}$ is corresponding predictive value from the model. For the nMSE, $\text{Y}_i$ is ground truth of the target at time point $i$ and $\hat{\text{Y}}_i$ is corresponding predictive value by the model. We reported the mean and standard deviation based on 20 iterations of testing on different data splits.

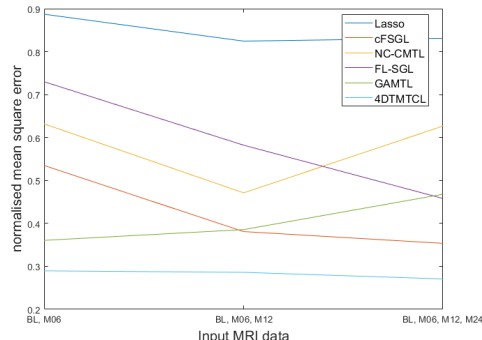 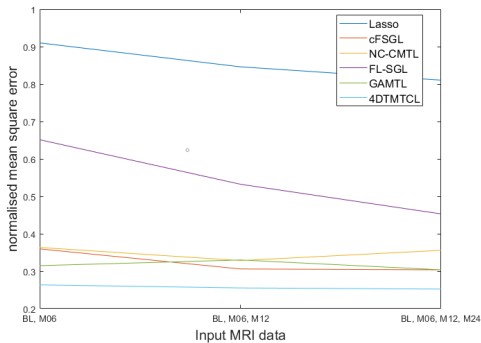

Figure 4: nMSE comparison for MMSE prediction.

Figure 5: nMSE comparison for ADAS-Cog prediction.

## 5 RESULTS AND DISCUSSION

### 5.1 COMPARISON WITH THE BENCHMARKS AND STATE-OF-THE-ARTS

We utilised the presented amalgamated magnitude-direction quantification to construct a tensor of brain structural variations, in conjunction with the proposed 4D tensor multi-task continual learning (4DTMTCL) algorithm, to compare with the following single task learning benchmarks and state-of-the-art MTL methods that were selected as competitive approaches in clinical deterioration prediction research. Including Lasso regression (Lasso) (Tibshirani, 1996), Convex Fused Sparse Group Lasso (cFSGL) (Zhou et al., 2013), Non-Convex Calibrated Multi-Task Learning (NC-CMTL) (Nie et al., 2018), Fused Laplacian Sparse Group Lasso (FL-SGL) (Liu et al., 2018) and Group Asymmetric Multi-Task Learning (GAMTL) (Oliveira et al., 2022). Tables 2 and 3, Figure 4 and 5 demonstrate the experimental results of MMSE and ADAS-Cog predictions respectively.

In terms of overall regression performance, our proposed approach outperforms single task learning, benchmarks and state-of-the-art MTL approaches in terms of nMSE for both MMSE and ADAS-Cog cognitive scores. Moreover, the proposed approach provides a lower rMSE than comparable alternatives for all single time points. Our main observations are as follows: 1) The proposed 4DTMTCL model outperforms single-task learning, benchmarks and state-of-the-art MTL models, demonstrating the utilisation of brain structural variation trend correlation calculations and biomarker latent factor hypothesis in our MTL formulation. 2) The proposed 4DTMTCL approach significantly improves prediction stability. The experimental results achieved through 20 iterations exhibited a lower standard deviation when referred to other comparable methods. This may be as a result of the biomarker latent factors that were incorporated into the prediction algorithm to enhance the stability. 3) In the scenario of AD dynamic prediction, the proposed 4DTMTCL model can achieve outstanding results. The knowledge from the previous models and the present model are combined in the present prediction model. Prediction accuracy can be increased by time-continuous MRI recordings of the participants, and when additional time-continuous MRI recordings are provided, prediction accuracy increases over time. Contrarily, the inclusion of time-continuous MRI recordings has no beneficial effect on the benchmarks and state-of-the-art competing approaches.

### 5.2 INTERPRETABILITY OF STRUCTURAL VARIATION CORRELATIONS BETWEEN BRAIN BIOMARKERS

In medical research, model performance is equally crucial as the methods' and results' interpretability. The cornerstone to current treatment is AD early detection and prevention since there is presently no cure for AD. Therefore, recognizing important biomarker structural variation correlations in early MRI data (in our study, the early MRI data relate to the BL and M06 consecutive data.) can aid clinicians in identifying patients with suspected AD for early prevention. Because the MMSE dataset is larger at each time point than the ADAS-Cog dataset, it provides a more extensive sample range. Tables 4, 5, 6, 7 and 8 exhibit the top ten brain biomarker correlations for the proposed 4DTMTCL model in decreasing order of weighted parameter values predicted by MMSE at various time points. Higher values indicate a greater influence on the final prediction. And they can be utilised as potential indicators for AD early detection.

Table 4: The top-10 rank brain biomarker correlations in time point M12 for 4DTMTCL approach on MMSE prediction.

| Brain biomarker correlation | Weight |
|---|---|
| Vol(C). of R.InferiorParietal - CTA. of R.InferiorParietal | 0.9764 |
| Vol(C). of R.Postcentral - CTA. of R.Postcentral | 0.9244 |
| Vol(C). of R.SuperiorParietal - Vol(C). of L.SuperiorParietal | 0.8652 |
| Vol(C). of R.Postcentral - CTA. of L.Postcentral | 0.7776 |
| Vol(C). of R.Postcentral - Vol(C). of R.RostralMiddleFrontal | 0.7540 |
| CTA. of L.Paracentral - Vol(C). of L.SuperiorParietal | 0.7535 |
| CTA. of R.SuperiorParietal - CTA. of R.InferiorParietal | 0.7526 |
| Vol(C). of R.Postcentral - CTA. of L.SuperiorParietal | 0.7390 |
| Vol(C). of R.Paracentral - Vol(C). of R.LateralOccipital | 0.7364 |
| Vol(WM). of L.LateralVentricle - Vol(WM). of R.LateralVentricle | 0.7283 |

Table 5: The top-10 rank brain biomarker correlations in time point M24 for 4DTMTCL approach on MMSE prediction.

| Brain biomarker correlation | Weight |
|---|---|
| Vol(C). of R.Postcentral - CTA. of R.Postcentral | 1.1057 |
| Vol(C). of R.InferiorParietal - CTA. of R.InferiorParietal | 1.0922 |
| Vol(C). of R.SuperiorParietal - Vol(C). of L.SuperiorParietal | 0.9003 |
| CTA. of R.SuperiorParietal - CTA. of R.InferiorParietal | 0.8696 |
| Vol(C). of R.Postcentral - CTA. of L.Postcentral | 0.8612 |
| Vol(C). of R.Postcentral - Vol(C). of R.RostralMiddleFrontal | 0.8392 |
| Vol(C). of R.Paracentral - Vol(C). of R.LateralOccipital | 0.8347 |
| Vol(C). of R.Precuneus - CTA. of R.RostralMiddleFrontal | 0.7864 |
| Vol(C). of R.Postcentral - CTA. of L.SuperiorParietal | 0.7801 |
| Vol(C). of R.Paracentral - CTA. of R.Postcentral | 0.7779 |

Table 6: The top-10 rank brain biomarker correlations in time point M36 for 4DTMTCL approach on MMSE prediction.

| Brain biomarker correlation | Weight |
|---|---|
| Vol(C). of R.Postcentral - CTA. of R.Postcentral | 1.2441 |
| Vol(C). of R.InferiorParietal - CTA. of R.InferiorParietal | 1.1859 |
| Vol(C). of R.SuperiorParietal - Vol(C). of L.SuperiorParietal | 1.1019 |
| Vol(C). of R.Postcentral - CTA. of L.Postcentral | 0.9752 |
| Vol(C). of R.Paracentral - Vol(C). of R.LateralOccipital | 0.9696 |
| CTA. of R.SuperiorParietal - CTA. of R.InferiorParietal | 0.9474 |
| Vol(C). of R.Paracentral - CTA. of R.Postcentral | 0.9384 |
| CTA. of L.Paracentral - Vol(C). of L.SuperiorParietal | 0.9141 |
| Vol(C). of R.Postcentral - Vol(C). of R.RostralMiddleFrontal | 0.9121 |
| Vol(C). of R.Postcentral - CTA. of L.SuperiorParietal | 0.8842 |

Table 7: The top-10 rank brain biomarker correlations in time point M48 for 4DTMTCL approach on MMSE prediction.

| Brain biomarker correlation | Weight |
|---|---|
| Vol(C). of R.Postcentral - CTA. of R.Postcentral | 1.3417 |
| Vol(C). of R.InferiorParietal - CTA. of R.InferiorParietal | 1.2118 |
| Vol(C). of R.SuperiorParietal - Vol(C). of L.SuperiorParietal | 1.1622 |
| Vol(C). of R.Paracentral - Vol(C). of R.LateralOccipital | 1.1226 |
| Vol(C). of R.Postcentral - CTA. of L.Postcentral | 1.0592 |
| Vol(C). of R.Postcentral - Vol(C). of R.RostralMiddleFrontal | 1.0019 |
| CTA. of R.SuperiorParietal - CTA. of R.InferiorParietal | 0.9962 |
| CTA. of L.Paracentral - Vol(C). of L.SuperiorParietal | 0.9735 |
| Vol(C). of R.Paracentral - CTA. of R.Postcentral | 0.9443 |
| Vol(C). of R.Postcentral - CTA. of L.SuperiorParietal | 0.9296 |

Table 8: The top-10 rank brain biomarker correlations in time point M60 for 4DTMTCL approach on MMSE prediction.

| Brain biomarker correlation | Weight |
|---|---|
| Vol(C). of R.InferiorParietal - CTA. of R.InferiorParietal | 1.4961 |
| Vol(C). of R.Postcentral - CTA. of R.Postcentral | 1.3820 |
| Vol(C). of R.SuperiorParietal - Vol(C). of L.SuperiorParietal | 1.2638 |
| Vol(C). of R.Paracentral - Vol(C). of R.LateralOccipital | 1.2409 |
| CTA. of L.Paracentral - Vol(C). of L.SuperiorParietal | 1.1817 |
| CTA. of R.SuperiorParietal - CTA. of R.InferiorParietal | 1.0901 |
| Vol(C). of R.Postcentral - CTA. of L.Postcentral | 1.0871 |
| Vol(C). of R.Paracentral - CTA. of R.Postcentral | 1.0680 |
| Vol(C). of R.Postcentral - Vol(C). of R.RostralMiddleFrontal | 1.0577 |
| Vol(C). of L.InferiorParietal - Vol(C). of L.SuperiorParietal | 1.0199 |

## 6  CONCLUSION

We propose a tensor multi-task continual learning approach for AD dynamic prediction scenarios to predict the AD progression at different time points in to simultaneously overcome the problems of monotonic data forms, small datasets and the scarcity of time-continuous data. In our approach, a multi-dimensional tensor-based predictive model is developed based on the correlation of the structural variation trends across brain biomarkers to address the monotonic data form problem, as well as the exploitation of tensor latent factors as multi-task relationships to share knowledge between patients to enhance model performance under small data set problems. AD dynamic prediction suffers from the problem of time-continuous data scarcity, which means that the number of available datasets decreases further as the disease progresses, and the proposed approach exploits the concept of continual learning to integrate time-continuous MRI recordings of patients in order to continuously improve the predictive accuracy of AD progression. The experimental results demonstrate that the proposed approach has the ability to diagnose and predict AD progression, that it has the capability to recognise brain structural variations in individuals with AD, MCI and CN, it only requires MRI data to achieve exceptional predictive performance, and that the model's performance improves as the number of MRI data increases.

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
