# OpenReview forum: "4D Tensor Multi-task Continual Learning for Disease Dynamic Prediction"
_ICLR.cc/2024/Conference — Submitted to ICLR 2024_

### Official Review · Reviewer_ZWzC · 2023-10-12

**Soundness:** 3 good
**Presentation:** 1 poor
**Contribution:** 2 fair
**Rating:** 5
**Confidence:** 2

**Summary:**

This work tackles prediction of Alzheimer's disease progression from small, scarce time-continuous and dynamic form of data. They propose 4D tensor multi-task continual learning algorithm, which utilises a tensor-based MTL algorithm to integrate and share spatio-temporal information. They evaluate this on the data from the Alzheimer’s Disease Neuroimaging Initiative (ADNI). The authors identified and analysed important relative structural variation correlations between brain biomarkers in the prediction of AD progression, which could be utilised as potential indicators for early identification of AD.

**Strengths:**

+ Very relevant research question.
+ The authors puts effort to interpret the results and understand the biomarkers.

**Weaknesses:**

- The paper is quite hard to follow. E.g the authors introduce in section 3.2 amalgamated magnitude-direction. Afterwards, for me it is puzzling to know what happens with that or where it is used. Because in section 3.3 the authors talk about the learning and regression problem.
- The methods section is also quite hard to follow with the equations. I would re-iterate or breakdown section 3.3 and eq. (2).
- Overall I feel lost in the details and I am missing the high-level info on how the data and task looks like.

**Questions:**

- How do you get the MRI brain biomarkers? Until the point of the biomarkers explained on p6 (which I guess are the features) the reader wonders what it is. Although it has been mentioned since p.4 section 3.2. So I would recommend the authors to reiterate this.
- Eq. 2: what is C_t? The description is missing
- Tables arrive earlier than mentioned, which makes it harder to follow. Please change it.
- What is the number of MRI scans? Short intro on ADNI would be helpful to follow.

---

> ### Author Response · Authors · 2023-11-15
>
> Thank you for your careful reading and detailed evaluation of our paper. Please find our response to your questions and concerns below.
>
> For Weaknesses 1: For the small dataset problem, the ADNI is the most commonly used data source and it was initiated in 2003, so it's 20 years from now, but according to the offical website, there are only 2760 participants. This is a relatively small amount of data in the field of computer science. After data pre-processing, the useful data for each research will usually become less than a thousand.  And the data elaboration in Table 1 demonstrates that the amount of AD data becomes lower and lower as time progresses. If our paper makes it to the revision stage, we will definitely add this information into the paper.
>
> For Weaknesses 2: In Figure 1 we show that dimension 1 and dimension 2 are both biomarker dimensions and their feature labels are the same, i.e. we are quantifying the intrinsic correlation between biomarkers and their ability to demonstrate globalised information about patient's brain structural variability.
>
> For Weaknesses 3: Thanks for the comment, we will definitely revise the Figure 2 if our paper can make it to the revision stage.
>
> For Question 1: Indeed in this paper, second order and third order are convertible to two-dimensional and three-dimensional, there is no significant difference between the two representations, the reason why we use the second order and third order terminology is that we want to explicitly state the number of subscripts for each element in the matrix and the tensor, the second-order matrix indicates that there are two subscripts for each element in the matrix, and the third-order tensor indicates that there are three subscripts for each element in the tensor. Thank you for your recommendation, if we can reach the revision stage for our paper, we will carefully consider which expression is more suitable for readers to understand and read.
>
> For Question 2: Thanks for the comment, we'll do more careful proofreading and paraphrasing, but it's important to point out that our team didn't use ChatGPT in the writing process.
>
> For Question 3: C is the latent factor matrix for the patient dimension, since it does not have a critical effect in the algorithmic operations and interpretability discussion of our proposed approach, it is only briefly mentioned in the optimisation process that it is a latent factor matrix. This is our mistake and we will definitely correct it as well, apologies for the confusion in your reading.
>
> Have a good day~ Kind regards

---

### Official Review · Reviewer_vwzZ · 2023-10-31

**Soundness:** 2 fair
**Presentation:** 2 fair
**Contribution:** 3 good
**Rating:** 6
**Confidence:** 4

**Summary:**

The authors claim that some challenges in predicting the progression of Alzheimer’s disease (AD) are monotonic data form, small dataset and scarcity of time-continuous data. To tackle these challenges, the authors propose a novel approach with 4D tensor multi-task continual learning. It is claimed that the proposed method integrates information from all available data and gets updated in a continual-learning fashion. The authors further argue that their method can achieve better accuracy and stability than single-task learning and SOTA multi-task regression methods in the prediction of several cognitive scores of AD progression.

**Strengths:**

1.	The idea to construct a 4D tensor representation of disease progression for multivariate spatiotemporal information aggregation is an intuitive idea for this particular task.
2.	The biomarker correlation analysis (Table 4 – 8) is quite thoughtful. With that said, it would have been better if more insights can be provided that relate these biomarkers and existing literature.

**Weaknesses:**

1.	In the second paragraphs of the Introduction section, the authors described the “three main problems” with existing models for AD progression. The first claimed problem is “data on neurological diseases such as AD are difficult to obtain”, but this claim sounds erroneous without additional context. It would be better if the authors specify the data modalities with limited availability or accessibility. Just as a reference, T1-weighted MRI data seems to be abundant --- I can name a few datasets with moderate-to-large scale with AD patients: the Alzheimer’s Disease Neuroimaging Initiative (ADNI) which the authors used in this paper, Anti-Amyloid Treatment in Asymptomatic Alzheimer’s (A4), and Open Access Series of Imaging Studies (OASIS).
2.	While I appreciate that the authors attempt to illustrate the 4D tensor data in Figure 1, under the current form it is still unclear how the first two dimensions are constructed. It seems like a 2D matrix, so what do the rows and columns represent? From the Introduction section it seems to be two distinct biomarkers, but suppose the matrix is $ M \times N $, what will the M and N feature dimensions represent? This seems a bit unclear from the figure.
3.	Figure 2 needs to be improved. The text over colored arrows is hard to read and looks unpleasing.

**Questions:**

1.	The authors seem to be using the terms “second-order matrix” and “third-order tensor” to refer to “2D matrix” and “3D tensor”. Would it be better to use “two-dimensional” and “three-dimensional” instead?
2.	There are a few grammar issues. I would recommend having additional rounds of proof-reading and paraphrasing. ChatGPT might be a valuable resource, though you may need to use with caution as it can easily change the meaning.
3.	In Section 3.3, where is $C$ defined?

---

> ### Author Response · Authors · 2023-11-15
>
> Thank you for your careful reading and detailed evaluation of our paper. Please find our response to your questions and concerns below.
>
> For Weaknesses 1: For the small dataset problem, the ADNI is the most commonly used data source and it was initiated in 2003, so it's 20 years from now, but according to the offical website, there are only 2760 participants. This is a relatively small amount of data in the field of computer science. After data pre-processing, the useful data for each research will usually become less than a thousand.  And the data elaboration in Table 1 demonstrates that the amount of AD data becomes lower and lower as time progresses. If our paper makes it to the revision stage, we will definitely add this information into the paper.
>
> For Weaknesses 2: In Figure 1 we show that dimension 1 and dimension 2 are both biomarker dimensions and their feature labels are the same, i.e. we are quantifying the intrinsic correlation between biomarkers and their ability to demonstrate globalised information about patient's brain structural variability.
>
> For Weaknesses 3: Thanks for the comment, we will definitely revise the Figure 2 if our paper can make it to the revision stage.
>
> For Question 1: Indeed in this paper, second order and third order are convertible to two-dimensional and three-dimensional, there is no significant difference between the two representations, the reason why we use the second order and third order terminology is that we want to explicitly state the number of subscripts for each element in the matrix and the tensor, the second-order matrix indicates that there are two subscripts for each element in the matrix, and the third-order tensor indicates that there are three subscripts for each element in the tensor. Thank you for your recommendation, if we can reach the revision stage for our paper, we will carefully consider which expression is more suitable for readers to understand and read.
>
> For Question 2: Thanks for the comment, we'll do more careful proofreading and paraphrasing, but it's important to point out that our team didn't use ChatGPT in the writing process.
>
> For Question 3: C is the latent factor matrix for the patient dimension, since it does not have a critical effect in the algorithmic operations and interpretability discussion of our proposed approach, it is only briefly mentioned in the optimisation process that it is a latent factor matrix. This is our mistake and we will definitely correct it as well, apologies for the confusion in your reading.
>
> Have a good day~ Kind regards

---

> ### Comment · Reviewer_vwzZ · 2023-11-21
> **Response to Rebuttal**
>
> Thanks for the rebuttal.
>
> A few points to make.
>
> 1. Weakness 1 is well addressed. The author gave a very good explanation.
> 2. Weakness 2 is still a concern, as I am still a bit confused even after reading the clarification.
> 3. Regarding the suggestion for grammar correction and proofreading --- I apologize for my ambiguous statements. I was just suggesting ChatGPT for proofreading because it's fast and convenient, which is allowed by this conference _"The use of LLMs is allowed as a general-purpose writing assist tool"_, with no other assumptions or judgements. But thanks for the clarification.
>
> For now, I will keep the current score. I can increase the rating if the authors can provide additional clarification on Weakness 2, through a more detailed and organized description of the biomarker dimensions. As I previously said, if the two biomarker dimensions form a matrix of size $M \times N$, I would like to know what the entry $[i, j]$ in that matrix represents.
>
> Thanks and best regards.

---

> > ### Author Response · Authors · 2023-11-22
> >
> > Thanks for the reply.
> >
> > For point 2: We apologise for the confusion in the description. For the 4D tensor in Figure 1, it consists of multiple 3D tensors in the time dimension. For each 3D tensor, it has three dimensions, which are dimension 1: biomarker dimension, dimension 2: biomarker dimension, and dimension 3: patient sample dimension.
> >
> > Both dimension 1 and dimension 2 are the same biomarker dimension with the same feature labels, and in the case for one of the 3D tensor, assuming that the number of patient dimensions is N and the number of biomarker dimensions is M, the 3D tensor has the dimensions M x M x N. The reason why dimension 1 and dimension 2 have the same features is because in this research we only want to quantify and investigate the intrinsic correlation between brain biomarkers, (e.g., when the structure of the hippocampus changes, how does the structure of the rest of the brain change), and to explore in our research whether this brain biomarker correlation can be utilised to differentiate between individuals at different levels of cognitive impairment as well as predicting the disease progression of individuals. If we are able to make it to the revision stage, we will make modifications in the paper regarding the explanation of dimensions to make the paper better readable and understandable.
> >
> > For point 3: Thank you for your advice and information, we apologise for misunderstanding your comment, we did miss this part of information about LLM as an assist tool on the official website.
> >
> > Thanks again for your valuable comments. Your comments are very helpful. Have a good day~ Kind regards

---

> > > ### Author Response · Authors · 2023-11-22
> > >
> > > Sorry this following reply was not entered successfully, each entry in the 3D tensor (M x M x N) represents a structural variation correlation between two brain biomarkers for one of the patient samples.

---

> > > ### Comment · Reviewer_vwzZ · 2023-11-23
> > > **Response to Rebuttal**
> > >
> > > I would like to thank the authors for the detailed clarification. I have increased my rating from 5 to 6.

---

> > > > ### Author Response · Authors · 2023-11-23
> > > >
> > > > We appreciate your patience in reviewing our responses, and glad we have clarified it clearly.
> > > >
> > > > Best wishes

---

### Official Review · Reviewer_6zjs · 2023-10-31

**Soundness:** 2 fair
**Presentation:** 2 fair
**Contribution:** 2 fair
**Rating:** 5
**Confidence:** 4

**Summary:**

The authors proposed an approach to address challenges in predicting the progression of Alzheimer’s disease (AD) due to issues with the data form, small dataset, and time-continuous data scarcity. The 4D tensor multi-task continual learning algorithm is used to quantify multi-dimensional information on brain structural variation and facilitate knowledge sharing between patients, continuously updating and optimizing prediction results as patient data increases. The proposed approach outperforms other methods in predicting AD progression using data from the Alzheimer’s Disease Neuroimaging Initiative and accurately identifies structural brain variations using magnetic resonance imaging (MRI) data alone.

**Strengths:**

+ Tackling the continual learning problem in medical longitudinal data

**Weaknesses:**

- It is not clear how the biomarkers are generated for the modeling.
- The proposed method is limited to structured data of biomarkers. It may not be generalized to other data format.
- The multi-task learning is not clearly defined and introduced in the presented work
- It is not clear how M12 ... M60 is composed. Are they overlapped with each other?
- what exactly is W_t in Eq. 2? What's the model parameter matrix as introduced?
- The dataset ADNI used in the experiments is not clearly introduced. What's the data split used? How many data samples are really used in the experiments?
- what is the trained disease progression predictive model? What's the model architecture?
- It will be helpful to see the performance of models trained for each task alone instead of one model for all the multi-tasks.
- The experimental setting and presented results seem to avoid the problem of "data form, small dataset, and time-continuous data scarcity" raised by the authors by introducing the structured data (unknown biomarkers), a single dataset (without details), a fixed longitudinal dataset with regular follow-up timescales (12:12:60, every 12 months)

**Questions:**

See Weaknesses

---

> ### Author Response · Authors · 2023-11-15
>
> Thank you for your careful reading and detailed evaluation of our paper. Please find our response to your questions and concerns below.
>
> For Question 1: In section 4.1 we illustrated the extraction of biomarker information from MRI utilising the FreeSurfer image analysis system, then utilise the amalgamated magnitude-direction quantification approach proposed in 3.2 to construct the extracted biomarker information into a three-dimensional tensor.
>
> For Question 2: The focus of this paper is indeed on biomarker structured data, but the proposed methodology can be applied to tabular data in any research direction, as long as the data permits. The methodology of this research is utilised to quantify correlations between features and constructed as a three-dimensional tensor, which can then be utilised to perform the downstream tasks of regression or classification. The quantification approach can be modified to suit the specific research direction, and the benefit of this process is that it is extremely strong in ensuring the interpretability of the model and results.
>
> For Question 3: We illustrated in the fourth paragraph of Section 1 that 'The predictions for each patient sample (the task in this research) share these latent factors', and then in the second paragraph of Section 2 we illustrated that 'we set up prediction task for a single patient as a task, which is a small-scale manner of task setting'.
>
> For Question 4: In the caption to Figure 2 we elaborated that 'The notation "BL" denotes the date of the patient's first admission for screening, "M06" denotes the time point 6 months after the first visit, "M12" denotes the time point 12 months after the first visit, etc.', M12 ... M60 are all the various time points of the disease,  they do not overlap with each other.
>
> For Question 5: W_t is model parameter matrix for the t-th prediction time point, introduced V is knowledge base matrix from continual learning which stores principal biomarker latent factors from all preceding model predictions.
>
> For Question 6: The ADNI dataset utilised is described in Section 4.1 and its the most commonly used data source in the field of AD research. The part of data split is described in section 4.2, where we used a ratio of 9:1 to divide the training and testing sets. The number of data samples used for the experiments is elaborated in Table 1.
>
> For Question 7: Our trained disease progression prediction model can receive patient MRI data and predict cognitive scores at multiple future time points to achieve the goal of disease progression prediction, and in the real world, patients with suspected AD will continue to go to the hospital for testing. Subsequent incremental data will be wasted if only a baseline model is used, or if the patient's serial testing records cannot be reasonably integrated. To address this problem, we applied the concept of continual learning to our approach, where predictions can be updated by allowing the model to receive new MRI data while receiving latent factor knowledge from all previous prediction models.
>
> For model architecture, a multi-dimensional tensor-based predictive model is developed based on the correlation of the  structural variation trends across brain biomarkers to address the monotonic data form problem, as  well as the exploitation of tensor latent factors as multi-task relationships to share knowledge between  patients to enhance model performance under small data set problems.  AD dynamic prediction  suffers from the problem of time-continuous data scarcity, which means that the number of available  datasets decreases further as the disease progresses, and the proposed approach exploits the concept  of continual learning to integrate time-continuous MRI recordings of patients in order to continuously  improve the predictive accuracy of AD progression.
>
> For Question 8: In this research, we set up prediction task for a single patient as a task, which is a small-scale manner of task setting, therefore it is less realistic to use the data from one of the patients alone for model training.
>
> For Question 9: The brain structure data is derived from MRI data, and we need to clarify that the data used in our experiments are indeed from the same data source, i.e., ADNI, which is the most commonly used data source in the field of AD research; however, we used two different datasets for the different cognitive scores tests (MMSE and ADAS-Cog), and the two cognitive scores testing procedures are also highly differentiated, which also demonstrates the performance generalisability of our proposed approach to different datasets.
>
> Have a good day~ Kind regards

---

> > ### Comment · Reviewer_6zjs · 2023-12-04
> > **rebuttal feedback**
> >
> > I thank the authors for the detailed response to my previous comments. Most of the unclarities are solved in the rebuttal, while they should be added to the new version. One thing that makes me feel the reported work is less convincing is the small amount of data employed. As mentioned in the rebuttal, the data set is too small to train separate models for each task, which will help demonstrate the benefit of the proposed method the most. Therefore, I would raise my rating; however, I am still leaning toward rejection.

---

### Official Review · Reviewer_Zsj5 · 2023-11-02

**Soundness:** 3 good
**Presentation:** 2 fair
**Contribution:** 2 fair
**Rating:** 5
**Confidence:** 4

**Summary:**

The manuscript proposes 4D Tensor Multi-Task Continual Learning to predict Alzheimer's disease progression. The method shows improved prediction performance across multiple time points compared to previous baselines.

**Strengths:**

- The manuscript compares the proposed approach with multiple baselines
- Interesting and exciting approach overall. However, I have a lot of questions.

**Weaknesses:**

As a continual learning algorithm:
- The experiments have been performed only with one neuroimaging dataset. Hence, the impact and empirical evidence are pretty limited, considering the general broader focus of the ICLR community. Furthermore, this experimental setup is relatively novel. Hence, it is tough to understand the significance of the proposed approach to the continual learning domain.
- it is unclear if the compared baselines apply well to the solved question. For example, in the survey of continual learning (Wang et al. 2023), there are eight continual learning scenarios. I think the manuscript has to be a bit more specific. While the related work only discussed Multi-Task Learning.

As a neuroimaging research:
- The biomarkers from Section 5.2 have not been checked with the literature. I do not see the hippocampus as usually damaged early compared to other regions (Rao et al., 2022).
- The related work for Alzheimer's and longitudinal studies is minimal and old (max up to 2013). For example, there exist more classical recent approaches (e.g., Marinescu et al., 2019). Also, ADNI was used for the TADPOLE challenge (Marinescu et al., 2018) with its leaderboard (https://tadpole.grand-challenge.org/Results/).

Wang, Liyuan, et al. "A comprehensive survey of continual learning: Theory, method and application." arXiv preprint arXiv:2302.00487 (2023).
Rao, Y. Lakshmisha, et al. "Hippocampus and its involvement in Alzheimer's disease: a review." 3 Biotech 12.2 (2022): 55.
Marinescu, Răzvan V., et al. "DIVE: A spatiotemporal progression model of brain pathology in neurodegenerative disorders." NeuroImage 192 (2019): 166-177.
Marinescu, Razvan V., et al. "TADPOLE challenge: prediction of longitudinal evolution in Alzheimer's disease." arXiv preprint arXiv:1805.03909 (2018).

**Questions:**

- Have you ensured that all the models have the same data available at each moment? Otherwise, the updated parameters in the proposed model will preserve the history, which might be unfair to the standard models learned only from the available data. How do you prepare features for the baselines? Do you treat new time points as additional features or different samples? Do you use scores from the previous time-point as input features to predict scores in the next time point? It will be great to clarify the experimental setup for the baselines. I also wonder if better feature engineering can achieve better performance with XGBoost / CatBoost Regression (instead of Lasso Regression).
- How does the algorithm scale computationally with the number of biomarkers?
- How many time points can the knowledge base preserve? Will the performance degrade over time and with respect to past?
- I do not see ablation for model parameters ($\lambda$, $\beta$ and $\theta$) and hyperparameters ($\alpha$ and $\eta$).
- The abstract claims that the model improves as the MRI data increases, but I do not see ablation for the training dataset size. But if you meant it increasing by having data from new time points, could it be just the case of having more data explaining the improved performance rather than an effect of continual learning?
- Figure 4 and Figure 5 do not show the variability of the approaches.
- Table 2 and Table 3 do not have a statistical comparison of the model's performances.

Wang, Liyuan, et al. "A comprehensive survey of continual learning: Theory, method and application." arXiv preprint arXiv:2302.00487 (2023).

---

> ### Author Response · Authors · 2023-11-15
>
> Thank you for your careful reading and detailed evaluation of our paper. Please find our response to your questions and concerns below.
>
> For Question 1: All models have the same feature data labels at each moment, but do not have the same number of samples. This is an significant problem in disease dynamic prediction, because the number of samples in progressive diseases will become smaller over time, not every patient will continue to go to the hospital or research institution for MRI examination. The approach we proposed is to enhance the accuracy, stability and interpretability of disease progression prediction in this disease dynamic prediction scenario.
>
> The input feature of the baseline comparison methods is a two-dimensional matrix (sample x feature). Since the construction of three-dimensional tensor data itself is a part of our proposed approach, the three-dimensional tensor data is not input to the baseline comparison methods.
>
> The feature data labels are the same for all time points, but the sample size is different, and the sample size becomes smaller and smaller as the disease progresses. The new time point is not an additional feature, because the sample size changes, the new time point has to re-train the model again, but the knowledge base of the trained model from the previous time points is integrated into the training process.
>
> The score at the previous time point is not used as an input feature to predict the score at the next time point. The proposed approach is utilised to predict the cognitive scores at each future time point in a single prediction. And the future cognitive score prediction is updated for the patient when new MRI data is available.
>
> The experimental setup of the baseline is basically the same as the experimental setup of our proposed approach, with the difference that it uses a 2D matrix as an input, whereas our proposed approach uses a 3D tensor as an input because the construction of the 3D tensor data is itself one of the parts of our proposed approach, and thus does not use the 3D tensor data as the basic experimental input data.
>
> The experiments in this paper are currently targeted to compare with single-task learning， benchmark and state-of-the-art multi-task learning methods in the disease dynamics prediction scenarios, therefore comparisons on ensemble learning methods are not performed at this time, but will be conducted in future researches to compare with a wider range of machine learning methods.
>
> For Question 2: The proposed approach can be computationally scaleable with varying number of biomarkers with great simplicity, which can be directly extended with the biomarker dimension of the 3D tensor construction method without the modification of other parts of the algorithm.
>
> For Question 3: Our knowledge base in this paper experiment preserves biomarker latent factor knowledge for 3 time periods (BL to M06, M06 to M12, M12 to M24), and from the experimental results in Tables 2 and 3, it can be demonstrated that the performance improves with the growth of the knowledge base (i.e., over time). Theoretically there is no upper limit to the amount of knowledge that can be preserved in a knowledge base if data and equipment permit.
>
> For Question 4: In the experiments of this paper, the model parameters and hyperparameters were selected by the cross-validation method.
>
> For Question 5: The change in the dataset size is demonstrated in Table 1 and it can be seen that the amount of data becomes smaller as time progresses, Tables 2 and 3 demonstrate that the performance of the model improves as time progresses and as the MRI data amount increases, and I must clarify that the amount of data does not increase but rather decreases because the model data for the next time period does not use the model data from the previous time period, the model for the next time period only utilises the knowledge base that has been constructed in the training process of all of the previous models.
>
> For Question 6: We wanted to clearly demonstrate the variation in the overall predictive performance of the proposed approach when the number of MRIs increases in Figures 4 and 5, therefore, we did not include the visualisation of variability to prevent the images from being too complex, however, your suggestion is absolutely valid, and if this paper can proceed to the revision stage, we will definitely include the visualisation of variability in the images.
>
> For Question 7: Figures 2 and 3 demonstrate the comparison of rMSE between the proposed approach and the comparative methods in terms of prediction performance at each individual time point, all values are the mean and standard deviation derived from 20 experiments. And Figures 4 and 5 demonstrate the comparison of nMSE between the proposed approach and the comparison methods in terms of overall regression performance.
>
> The references provided are all very exploitative, especially Wang et al. (2023).
>
> Have a good day~ Kind regards

---

> ### Author Response · Authors · 2023-11-15
> **For Weaknesses**
>
> For Weaknesses 1: We need to clarify that the data used in our experiments are indeed from the same data source, i.e., ADNI, which is the most commonly used data source in the field of AD research; however, we used two different datasets for the different cognitive scores tests (MMSE and ADAS-Cog), and the two cognitive scores testing procedures are also highly differentiated, which also demonstrates the performance generalisability of our proposed approach to different datasets.
>
> The main intention of this research is to demonstrate the effectiveness of continual learning integrated with tensor multi-task learning in the problem of disease dynamics prediction
>
> For Weaknesses 2 and 4: The references provided are all very exploitative, especially Wang, Liyuan, et al. "A comprehensive survey of continual learning: theory, method and application." arXiv preprint arXiv. 2302.00487 (2023). If our paper makes it to the revision stage, we will definitely update the related work chapter. But we need to point out that the experimental comparison methods we have chosen are benchmark and SOTA highly cited papers published in top conferences and top journals in the relevant fields.
>
> For Weaknesses 3: Current biomarker interpretability research focuses on the importance of individual biomarkers, whereas our research focuses on the impact of biomarker correlation on the AD progression, and there is a large gap between them, and it is not feasible to check our interpretability results from past studies, and the validity of the biomarker correlations identified in this research will require more in-depth research with clinicians and medical researchers.

---

> ### Comment · Reviewer_Zsj5 · 2023-11-22
> **Response to Authors**
>
> Thank you for the effort to address my concerns and questions. I decided to keep the score.
>
> > For Weaknesses 3: Current biomarker interpretability research focuses on the importance of individual biomarkers, whereas our research focuses on the impact of biomarker correlation on the AD progression, and there is a large gap between them, and it is not feasible to check our interpretability results from past studies, and the validity of the biomarker correlations identified in this research will require more in-depth research with clinicians and medical researchers.
>
> This response does not address my concern.
> Based on the method section of the paper, your correlation indicates the following ": 1) both grow, 2) both decline, 3) one grows, and the other declines, 4) one changes and the other does not change, and 5) both remain unchanged". These values seem interpretable; hence, you can verify whether your pairs are related to the literature, especially volume pairs like Vol(C) - Vol(C), which should address the structural variation. Furthermore, I do not see Hippocampal areas.
>
> > For Weaknesses 1: We need to clarify that the data used in our experiments are indeed from the same data source, i.e., ADNI, which is the most commonly used data source in the field of AD research; however, we used two different datasets for the different cognitive scores tests (MMSE and ADAS-Cog), and the two cognitive scores testing procedures are also highly differentiated, which also demonstrates the performance generalisability of our proposed approach to different datasets.
>
> This response does not address my concern because these are two different targets/tasks but not the datasets.
>
> > The references provided are all very exploitative, especially Wang et al. (2023).
>
> To clarify the ambiguity, the references are not "all very exploitative." Because the related work is poorly written. (A) it is written like nobody has researched AD in recent years, and (B) you have to discuss the continual learning work if you propose continual learning. However, the references are not my primary concern, and my references are given as the point of interest and to be more specific in your language. Furthermore, I do not have any association with these papers.
>
> >  For Question 6: We wanted to clearly demonstrate the variation in the overall predictive performance of the proposed approach when the number of MRIs increases in Figures 4 and 5, therefore, we did not include the visualisation of variability to prevent the images from being too complex, however, your suggestion is absolutely valid, and if this paper can proceed to the revision stage, we will definitely include the visualisation of variability in the images.
>
> This has to be done in the revision stage. If you have a concern with the clarity of visualization, you can add them to the appendix.
>
> > Table 2 and Table 3 do not have a statistical comparison of the model's performances.
>
> It has not been addressed. To ensure rigor, you need to statistically compare the best-performing model with others using Wilcoxon or T-test (if normality assumptions pass) with correct for multiple comparisons (e.g., Holm).

---

> > ### Author Response · Authors · 2023-11-22
> >
> > Thank you for your reply, regardless of the final score, we would like to address your concerns as well as provide clarification for our paper.
> >
> > For Weaknesses 3: Our biomarker correlation quantification approach is indeed interpretable, but its can simultaneously quantify information about the magnitude and direction of structural variation between biomarkers, and to the best of our knowledge, such information and knowledge is not implicated in current biomarker interpretability studies, which is why there is no hippocampus, and it is possible that we may have discovered or implicated new knowledge about biomarkers, which will require more in-depth research with clinicians and medical researchers in the future.
> >
> > For Weaknesses 1: Although both datasets use MRI data, the patient samples for the MMSE and ADAS-Cog datasets are not the same according to the ADNI official website, and the cognitive score ranges for the MMSE and ADAS-Cog are significantly different (the range of MMSE cognitive scores is from 0-30, and the range of ADAS-Cog cognitive scores is from 0-70), which proves the generalisability of the performance by our proposed approach for different datasets.
> >
> > For related work concerns: What we mean in the related work section is that the field of AD research is not currently fully developed in brain biomarker relationships and is not utilising this knowledge within machine learning algorithms to improve the performance of AD progression prediction. For continual learning, the original intention of this research was to utilise it as an applied approach to solve the novel problem of disease dynamics prediction, and we are aware that if viewed in isolation as a methodology for continual learning, it cannot be considered as a novel methodology because the topic of this research is not about continual learning, but rather about how to solve disease dynamics prediction along with its series of derived sub-problems. If this paper can progress to the revision stage, we will certainly consider adding information about continual learning related work to increase the readability and comprehensibility of the paper. In addition, we apologise if the language of our response has caused you some misunderstanding.
> >
> > For Question 6: Thanks for the comment, we will definitely complete it in revision if we have chance.
> >
> > For Table 2 and Table 3: In the experiments of this research, we demonstrate in Tables 2 and 3 statistical comparisons of model performance at individual time points with rMSE values, and in Figures 4 and 5 statistical comparisons of overall model performance with nMSE values.
> >
> > Have a good day~ Kind regards

---

> > > ### Comment · Reviewer_Zsj5 · 2023-11-22
> > > **Response to Authors**
> > >
> > > > For Weaknesses 1: Although both datasets use MRI data, the patient samples for the MMSE and ADAS-Cog datasets are not the same according to the ADNI official website, and the cognitive score ranges for the MMSE and ADAS-Cog are significantly different (the range of MMSE cognitive scores is from 0-30, and the range of ADAS-Cog cognitive scores is from 0-70), which proves the generalisability of the performance by our proposed approach for different datasets.
> > >
> > > The scores are considered as different tasks, but can not be considered as different datasets (e.g., OASIS). Because the input distribution is the same, while the target distribution is different.
> > >
> > > > For Table 2 and Table 3: In the experiments of this research, we demonstrate in Tables 2 and 3 statistical comparisons of model performance at individual time points with rMSE values, and in Figures 4 and 5 statistical comparisons of overall model performance with nMSE values.
> > >
> > > My question is not to change the way how you evaluate the model, but to add additional rigor to your results. The statistical comparisons (e.g., with Wilcoxon) have to be performed on the scores of rMSE or nMSE. This is needed to ensure that there are significant differences in the performance of the models. Because model performance can be described as a mean and additional variability term across seeds or folds with standard deviation, standard error, or IQR. Due to the variability of the models you might have situations where the models are comparable. To verify the results one can perform pairwise statistical analysis between the performance scores of the models.
> > >
> > > > For Weaknesses 3: Our biomarker correlation quantification approach is indeed interpretable, but its can simultaneously quantify information about the magnitude and direction of structural variation between biomarkers, and to the best of our knowledge, such information and knowledge is not implicated in current biomarker interpretability studies, which is why there is no hippocampus, and it is possible that we may have discovered or implicated new knowledge about biomarkers, which will require more in-depth research with clinicians and medical researchers in the future.
> > > > For related work concerns: What we mean in the related work section is that the field of AD research is not currently fully developed in brain biomarker relationships and is not utilising this knowledge within machine learning algorithms to improve the performance of AD progression prediction. For continual learning, the original intention of this research was to utilise it as an applied approach to solve the novel problem of disease dynamics prediction, and we are aware that if viewed in isolation as a methodology for continual learning, it cannot be considered as a novel methodology because the topic of this research is not about continual learning, but rather about how to solve disease dynamics prediction along with its series of derived sub-problems. If this paper can progress to the revision stage, we will certainly consider adding information about continual learning related work to increase the readability and comprehensibility of the paper. In addition, we apologise if the language of our response has caused you some misunderstanding.
> > >
> > > Considering these two, the main contribution of this work is an application to neuroscience & cognitive science. Hence, the interpretability and empirical evidence on multiple datasets play a major role in assessing the significance of this work.

---

> > > > ### Author Response · Authors · 2023-11-23
> > > >
> > > > Thanks for the comments. Your comments are very helpful.
> > > >
> > > > For Weaknesses 1: Although both the MMSE and ADAS-Cog datasets come from the ADNI database, according to the ADNI official website, the samples of these two datasets are not exactly the same, which means that their input distributions are different. We will definitely utilise datasets from multiple databases in future research to enhance the comprehensibility of our research and address concerns in this aspect.
> > > >
> > > > For Table 2 and Table 3: Thanks for the comment. We will definitely consider utilising additional statistical methods in future research to include extra rigour.
> > > >
> > > > For Weaknesses 3: Our main focus in this work is indeed to address the disease dynamics prediction, which is a critical problem in neuroscience and cognitive science. We will carefully consider your comment to use multiple datasets in the future and to conduct more in-depth research on the interpretability of the proposed approach and experimental results in terms of biomarker function and clinical impact.
> > > >
> > > > Kind regards

---

### Author Response · Authors · 2023-11-15

We would like to thank all the reviewers for their comments and for recognising the novelty of our research, the importance of the problems addressed and the approach proposed, and the validity of the results. Below is a summary of the important contributions to our paper, followed by a response to each reviewer to resolve their concerns and provide clarification where necessary.

Primary contributions: We proposed a 4D tensor multi-task continual learning approach for AD dynamic prediction scenarios to predict the AD progression at different time points in to simultaneously overcome the problems of monotonic data forms, small datasets and the scarcity of time-continuous data.  The experimental results demonstrate that the proposed approach has the ability to diagnose and predict AD progression, that it has the capability to recognise brain structural variations in individuals with AD, MCI and CN, it only requires MRI data to achieve exceptional predictive performance, and that the model's performance improves as the number of MRI data increases. And we identified and analysed important relative structural variation correlations between brain biomarkers in the prediction of AD progression, which could be utilised as potential indicators for early identification of AD.

---

### Meta-Review · Area_Chair_FLmw · 2023-12-08

**Metareview:**

This submission contributes a continual-learning model to predict Alzheimer's disease progression from longitudianl MRI data. The submission generated excanges with the reviewers. The reviewers found the overall work promising. However, it is not clear that in its current state it meets to high bar for ICLR. Indeed, the reviewers were not fully convinced by the score and empirical validation, noting that they are fairly restricted, with only one application to only one dataset (of fairly limited size compared to typical continual learning settings.

**Justification For Why Not Higher Score:**

The sophistication of the approach does not match the fact that the dataset on which it is applied is fairly limited.

**Justification For Why Not Lower Score:**

N/A

---

### Decision · Program_Chairs · 2024-01-16

Reject